# Quantum wake dynamics in Heisenberg antiferromagnetic chains

A. Scheie [1] ✉, P. Laurell [2,3], B. Lake [4,5], S. E. Nagler [1,6], M. B. Stone [1], J-S Caux [7] & D. A. Tennant [1,6,8]

Traditional spectroscopy, by its very nature, characterizes physical system properties in the momentum and frequency domains. However, the most interesting and potentially practically useful quantum many-body effects emerge from local, short-time correlations. Here, using inelastic neutron scattering and methods of integrability, we experimentally observe and theoretically describe a local, coherent, long-lived, quasiperiodically oscillating magnetic state emerging out of the distillation of propagating excitations following a local quantum quench in a Heisenberg antiferromagnetic chain. This "quantum wake" displays similarities to Floquet states, discrete time crystals and nonlinear Luttinger liquids. We also show how this technique reveals the non-commutativity of spin operators, and is thus a model-agnostic measure of a magnetic system's "quantumness."

Ever since its introduction, the Heisenberg chain[1]

$$\mathcal{H} = J \sum_{i=1}^{N} \vec{S}_i \cdot \vec{S}_{i+1} \qquad (1)$$

has been the paradigmatic model of strongly correlated many-body quantum physics. Its exact solution by Bethe[2] gave birth to the field of quantum integrability; its magnetic excitations, spin-1/2 spinons[3], are the prototypical fractionalized excitations. The model is not simply a theoretical archetype, but also effectively describes many physical quantum magnets such as $KCuF_3$[4,5], in which the chains are formed by magnetic $Cu^{2+}$ ions hybridizing along the $c$ axis. Although $KCuF_3$ orders magnetically at $T_n = 39$ K, even below the ordering temperature its high energy spectrum retains the characteristic spinon spectrum[6] while exhibiting strong quantum entanglement[7].

One of the best experimental tools for studying magnetic excitations is inelastic neutron scattering[8], which measures the energy-resolved Fourier transform of the space- and time-dependent spin-spin correlation function $G(r,t) = \langle S_i^\alpha(0) S_{i+r}^\alpha(t) \rangle$, $(\alpha = x,y,z)$[9]. Accordingly, scattering cross section data is typically reported in terms of reciprocal space and energy. However, as pointed out by Van Hove in 1954[10,11], with enough data one can take the inverse Fourier transform and obtain the spin correlations in real space and time with atomic spatial resolution and time resolution of ~$10^{-14}$ s. This transformation was shortly thereafter applied to liquid lead neutron scattering data[12], and more recently on water using inelastic X-ray scattering[13] but has not been applied to magnetic materials.

Space-time dynamics in one dimension has been the subject of extensive study in recent decades[14], with attention mostly focusing on ballistically propagating excitations (describable using bosonization/ Luttinger liquid theory[15]) forcing "light-cone"-induced bounds on velocity of correlations and entanglement spreading[16]. The physics of Heisenberg chains is however much richer, containing nonlinearities whose effects can be captured exactly using integrability, or asymptotically using nonlinear Luttinger liquid theory[17].

In this paper, we use high-precision INS data transformed back to real, atomic-level space and time to characterize magnetic dynamics at the local level in a Heisenberg chain. We firstly show that this technique

[1]Neutron Scattering Division, Oak Ridge National Laboratory, Oak Ridge, TN 37831, USA. [2]Computational Sciences and Engineering Division, Oak Ridge National Laboratory, Oak Ridge, TN 37831, USA. [3]Department of Physics and Astronomy, University of Tennessee, Knoxville, TN 37996, USA. [4]Helmholtz-Zentrum Berlin für Materialien und Energie GmbH, Hahn-Meitner Platz 1, D-14109 Berlin, Germany. [5]Institut für Festkörperphysik, Technische Universität Berlin, Hardenbergstraße 36, D-10623 Berlin, Germany. [6]Quantum Science Center, Oak Ridge National Laboratory, Oak Ridge, TN 37831, USA. [7]Institute of Physics and Institute for Theoretical Physics, University of Amsterdam, PO Box 94485, 1090 GL Amsterdam, The Netherlands. [8]Shull Wollan Center - A Joint Institute for Neutron Sciences, Oak Ridge National Laboratory, Oak Ridge, TN 37831, USA. ✉e-mail: scheie@lanl.gov

provides access to spin-spin commutators, and is a measure of quantum coherence. We then focus on previously overlooked features of the real-space/time magnetic Van Hove correlation function $G(r,t)$, namely the effects of long-term coherent, non-propagating excitations (beyond the reach of bosonization). We observe a correlated time-dependent state resulting from the integrability-induced "persistent memory" of the Heisenberg chain. This state is reminiscent of a local many-body Floquet state or a discrete time crystal, in that it displays a characteristic time-repeating pattern with fixed period. Its correlations also display a remarkable (spatial) "period doubling" (mirroring the time period doubling of a discrete time crystal), in that the original site-alternating Néel order of the initial state changes to a two-site-spaced, oscillating antiferromagnetic correlation. This state, which we call a "quantum wake" due to its similarity to the wake created by a moving ship, is a coherent wavepacket of "deep" and "edge" spinons stabilized and made observable via a Van Hove singularity, and recalls the quantum dynamical impurity picture of nonlinear Luttinger liquid theory.

## Results and discussion

Experimental $G(r,t)$ results are obtained using available KCuF$_3$ data from Refs. 5, 18 (full details are provided in the "Methods" section).

The result is shown in Fig. 1, where ferromagnetic $G(r,t)$ correlations are shown in red and antiferromagnetic correlations are shown in blue. To help interpret the experimental $G(r,t)$, we also calculated $G(r,t)$ from: (i) Bethe Ansatz[5] for zero temperature, and (ii) semi-classical linear spin wave theory (LSWT). These are shown in Fig. 2. Experimental resolution broadening does affect the long-time $G(r,t)$ dynamics (see Supplementary Note 5), but the short-time spin correlations $G(r,t)$ are surprisingly insensitive to experimental resolution effects.

Real space $G(r,t)$ for spin systems can also be probed with cold atom and trapped ion experiments[19–21], but $G(r,t)$ derived from neutron scattering has several unique advantages: (i) The systems probed by neutrons are thermodynamic, and temperature is a well-defined quantity. (ii) Neutrons explore the spin system's evolution following a local perturbation. (iii) As we show below, neutron scattering accesses the imaginary $G(r,t)$ which reveals quantum coherence and Heisenberg uncertainty.

The Fourier transform of the $S(Q,\omega)$ scattering data produces a $G(r,t)$ with complex values, with a distinct interpretation for the real and imaginary parts. As noted by Van Hove[11], the imaginary part quantifies the imbalance between positive and negative energy

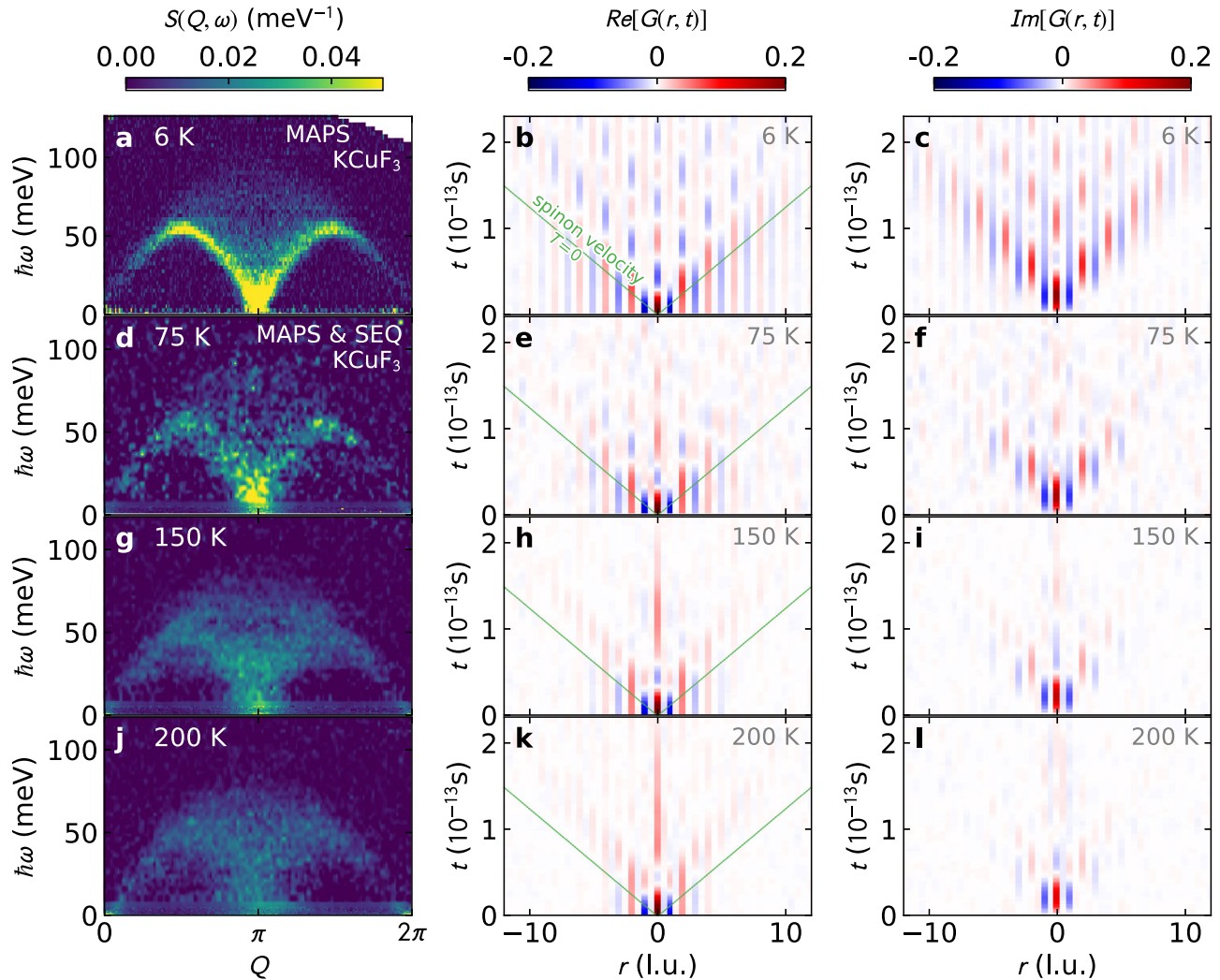

**Fig. 1 | Scattering and Van Hove correlations.** Finite temperature neutron scattering data for KCuF$_3$ from the MAPS and SEQUOIA (SEQ) instruments (left column: **a**, **d**, **g**, **j**) and their transformation to real-space correlations, with the real $G(r,t)$ (center column: **b**, **e**, **h**, **k**) and imaginary $G(r,t)$ (right column: **c**, **f**, **i**, **l**). Red indicates ferromagnetic spin correlation, blue indicates antiferromagnetic spin correlation. At low temperatures, the real $G(r,t)$ wavefront at the light cone is antiferromagnetic, and by 200 K it becomes ferromagnetic. Meanwhile, the imaginary $G(r,t)$ is restricted in space and time at higher temperatures, showing loss of quantum coherence.

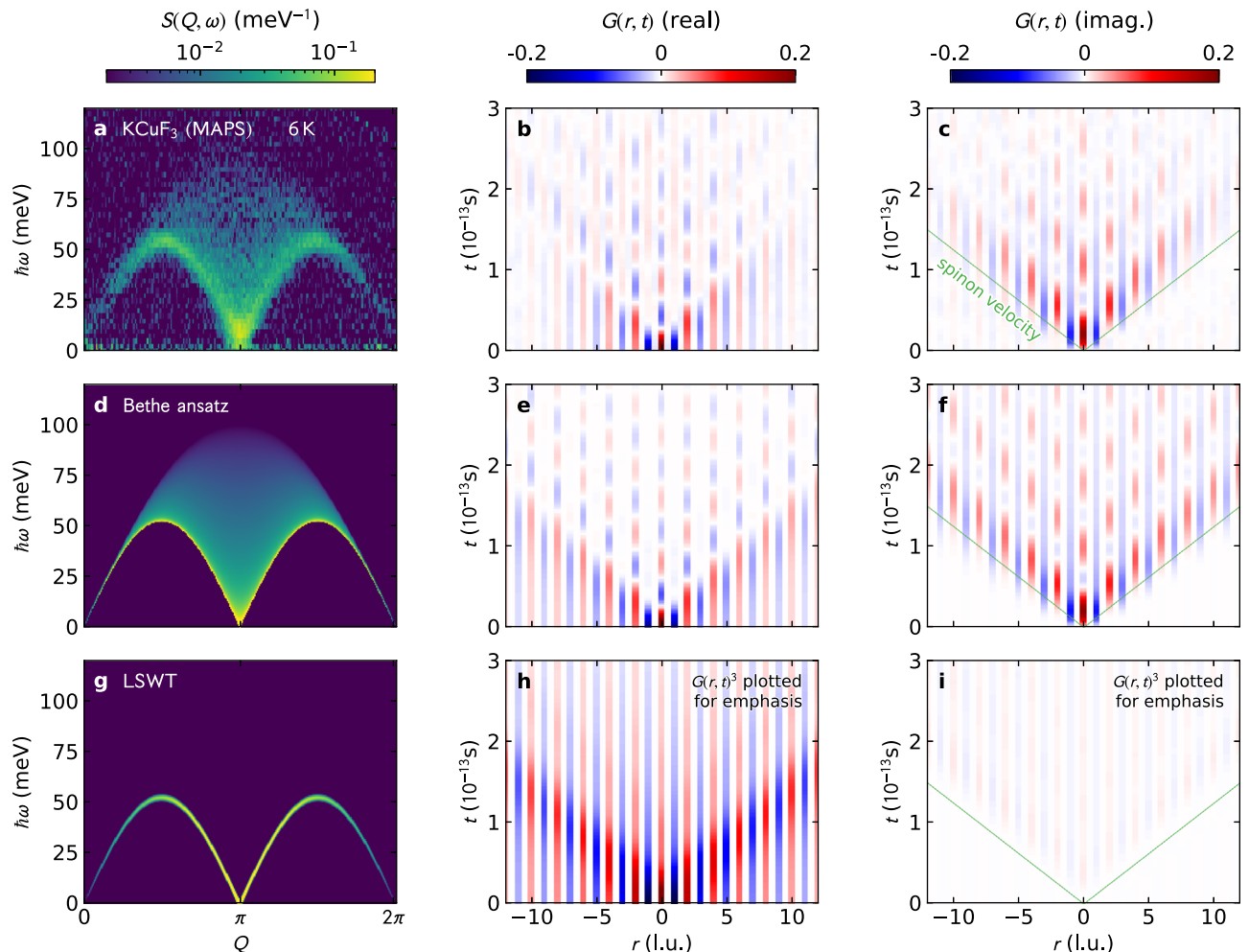

**Fig. 2 | Van Hove time-dependent real-space spin-spin correlation compared to theory with imaginary components.** **a** 6 K KCuF$_3$ scattering from the MAPS instrument, **b** real component of $G(r, t)$, **c** imaginary component of $G(r, t)$. Panels **d**–**f** show the same for $T = 0$ Bethe ansatz, and **g**–**i** show the same for $T = 0$ linear spin wave theory (LSWT) on a $S = 1/2$ HAF chain (renormalized by $\pi/2$ to match the light cone velocity in the top two panels). The thin green lines on $G(r, t)$ plots show the magnon/spinon velocity.

scattering, which when Fourier transformed becomes

$$Im[G(r,t)] = \frac{1}{2i}\left[ \langle S_i^z(0)S_j^z(t)\rangle - \langle S_i^z(0)S_j^z(t)\rangle^* \right],\qquad(2)$$

where $r$ is the distance between spins at sites $i$ and $j$. This can be rewritten with a commutator

$$Im[G(r,t)] = \frac{1}{2i}\langle [S_i^z(0), S_j^z(t)]\rangle.\qquad(3)$$

Following the same derivation, we arrive at the equation for the real part of $G(r, t)$

$$Re[G(r,t)] = \frac{1}{2}\langle \{S_i^z(0), S_j^z(t)\}\rangle.\qquad(4)$$

By Robertson's relation[22], a nonzero commutator between observables implies Heisenberg uncertainty; thus nonzero imaginary $G(r, t)$ indicates the presence of an uncertainty relation between $S_i^z(0)$ and $S_j^z(t)$. This mutual incompatibility is thus an indicator of quantum coherence between spins (see Supplementary Note 1). It is striking that the quantum coherence can be tracked as a function of temperature with the imaginary $G(r, t)$ in Fig. 1. As temperature increases,

the nonzero imaginary $G(r, t)$ shrinks to shorter and shorter times and distances, showing how the finite-temperature classical world emerges from the quantum world. On the other hand, the real part $Re[G(r, t)] = \frac{1}{2}\langle \{S_i^\alpha(0), S_{i+r}^\alpha(t)\}\rangle$ extracts classical behavior surviving even at infinite temperature.

The real space correlations in Figs. 1 and 2 emerge from a flipped spin at $t = 0$, $r = 0$. A number of things can be observed from these $G(r, t)$ data: first, the characteristic "light cone" defined by the spinon velocity $v = \frac{\pi J}{2}$ where $J$ is the exchange interaction. At low temperatures, everything below the light cone is static while everything above it is dynamic. Second, at low temperature in $G(r, t)$ there is a clear distinction between even and odd sites: the odd neighbor correlations quickly decay to zero above the light cone, whereas the even neighbor correlations persist to long times. Third, as temperature increases the spin oscillations above the light cone shrink to shorter distances and times, until by 200 K the on-site ($r = 0$) correlation oscillates only once and no neighbor-site oscillations are visible. Fourth and finally, the wavefront above the light cone changes to ferromagnetic at high temperatures (Fig. 1h) whereas it was antiferromagnetic at low temperatures. This accompanies the nonzero imaginary $G(r, t)$ shrinking to shorter and shorter times and distances as temperature increases.

To gain a better understanding of the signal, we should identify which excitations are responsible for which part of the $G(r, t)$ signal. The light cone is due to the low-energy correlations around $Q \sim \pi$ which

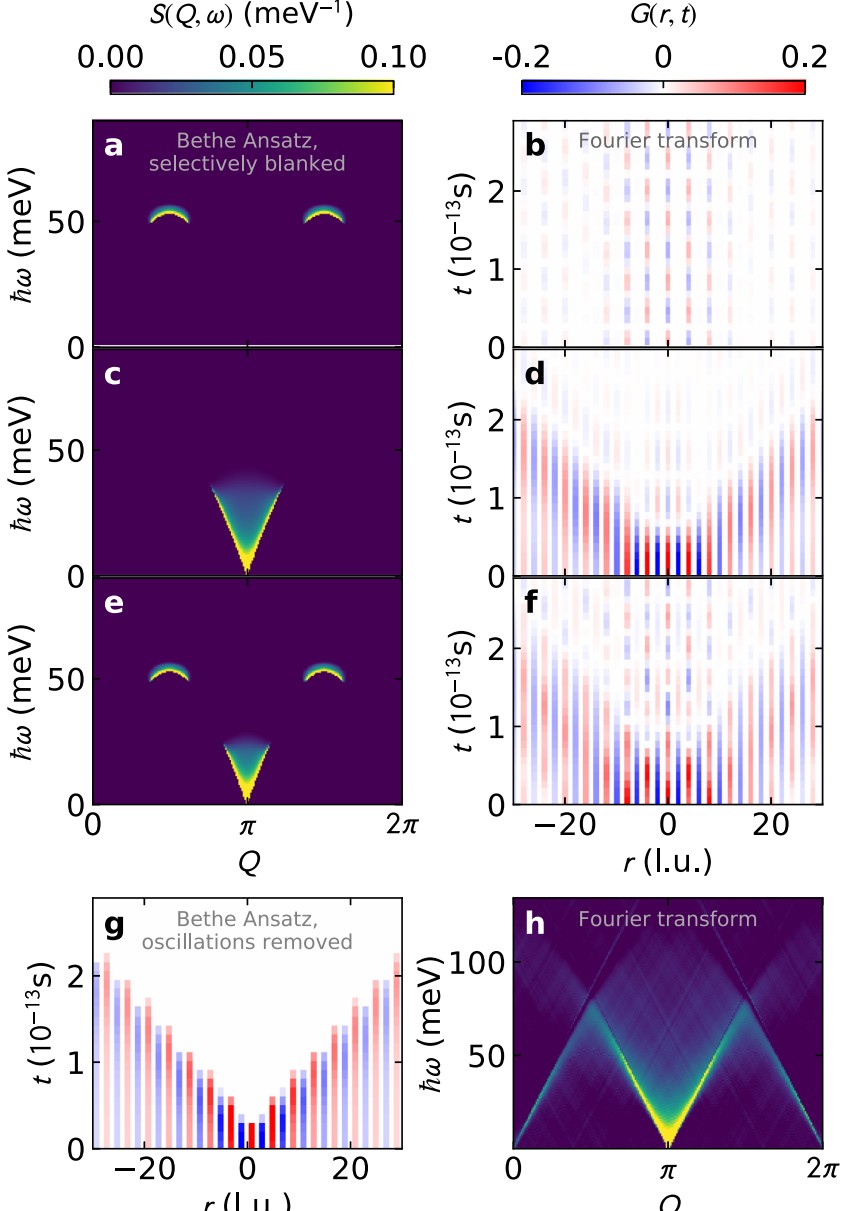

**Fig. 3 | Signal analysis of the Bethe Ansatz.** The right column is the Fourier Transform of the left. **a, c, e** The Bethe Ansatz with everything removed but key features at $Q = \pi$ or $Q = \pi/2$. **b, d, f** The resulting Fourier transform of these spectra into real space and time. This clearly shows that the oscillations above the light cone are due to the stationary $Q = \pi/2$ states, while the light cone is due to the dispersive $Q = \pi$ state. **g** The $G(r, t)$ of the Bethe Ansatz with all correlations above the light cone set to zero. Fourier-transforming this back into $S(Q, \omega)$ in **h**, we find a spinon spectrum with the $Q = \pi/2$ stationary states missing---confirming that these are responsible for the oscillating Floquet dynamics.

can be understood from traditional bosonization, the Fermi velocity being given by the group velocity of $Q$-$\pi$ spinons. These being the fastest-moving ballistic particles, they limit the velocity of energy, correlations, and entanglement propagation, giving the Lieb-Robinson bound[16,23]. Such a light cone is seen in theoretical simulations[24–29] and cold-atom experiments[20], and nicely also here in KCuF$_3$.

Letting the fast-moving ballistic particles "distill" away leaves a "quantum wake" behind the wavefront, a persistent oscillating state above the light cone which is clearly seen in Fig. 1 panel b and Fig. 2 panels b, c, e and f. This originates from another crucial characteristic of $S(Q, \omega)$, namely that its correlation weight is spread nontrivially within the spinon continuum. Contrasting LSWT with Bethe Ansatz in the second and third row of Fig. 2 shows stark differences in dephasing behavior. LSWT, being inherently coherent, has very slow dephasing and no quantum wake. For the experimental and Bethe Ansatz $G(r, t)$

however, there exist pockets of states around $Q$-$\pi/2$, $3\pi/2$, $\omega \simeq \pi J/2$ which display a Van Hove singularity in their density of states. Since the existence and sharpness of the lower edge are contingent on integrability, measuring the (slowness of the) time decay of the quantum wake is in fact a direct experimental measurement of the proximity to integrability.

To more illustratively map the features in $G(r, t)$ with specific spinon states, we selectively remove parts of the Bethe Ansatz $S(Q, \omega)$ spectrum, keeping only key features, and Fourier transform into $G(r, t)$. As shown in Fig. 3a, b, the oscillations above the light cone come from the $Q = \pi/2$ Van Hove singularities at the top of the spinon dispersion where the spinons have zero group velocity. Meanwhile, Fig. 3c, d shows the light cone emerges from the strongly dispersing low-energy $Q = \pi$ states. Combining these two states in Fig. 3e, f gives a rough reproduction of the actual $G(r, t)$, indicating that the $Q = \pi/2$ and $Q = \pi$

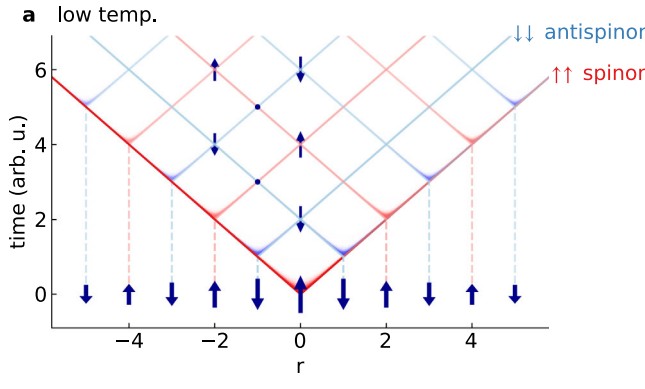

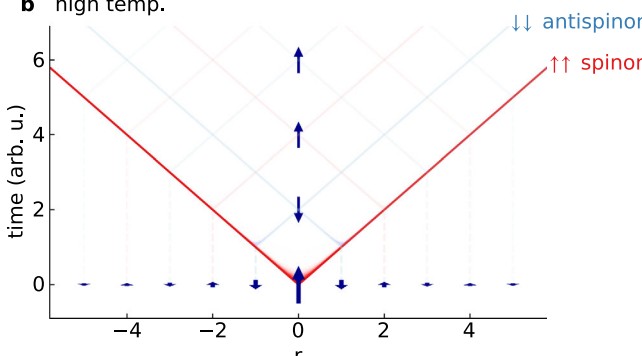

**Fig. 4 | Schematic description of the AFM Van Hove correlations.** At low temperatures, **a** a central spinon light cone emanates from $r = 0$, $t = 0$. As it reaches each neighboring site, it excites a pair of (red) spinons which creates its own light cone. Odd neighbor sites have opposite spin from $r = 0$ at $t = 0$, and thus they create (blue) antispinon pairs. Light cone shading indicates the strength/coherence of the quantum correlations. For even $r$, these spinon light cones create constructive interference and continue to flip spins up and down. For odd $r$, the spinons and antispinons destructively interfere, such that the correlations quickly go to zero. At high temperatures **b**, the spin correlations are much weaker, such that the spinon and antispinon light cones emanating from $|r| > 0$ are weakly coherent with $r = 0$ and thus their influence is suppressed, leading to oscillations restricted in both space in time as seen in Fig. 1.

spinon states are what give the Heisenberg chain quantum wake its distinctive properties. Bolstering this conclusion is the analysis shown in Fig. 3g, h where we remove the oscillations above the light cone from $G(r, t)$, and transform back into $S(Q, \omega)$. In this case, we see the familiar spinon spectrum, but with the stationary $Q = \pi/2$ states missing—showing that the flat singularity at the top of the spinon dispersion is responsible for the long-lived oscillating spin correlations.

Quantum scrambling: Perhaps the most striking feature of the KCuF$_3$ quantum wake is the total loss of Néel correlations above the light cone. Below the light cone, the system shows static $Q = \pi$ antiferromagnetism. Above the light cone, the system shows dynamic period-doubled $Q = \pi/2$ antiferromagnetism, with hardly a trace of the original state. In stark contrast to this, equal-time real space correlators $\langle S_i^\alpha(t)S_j^\alpha(t)\rangle$ (as opposed to dynamical correlator $G(r, t)$ which measures $\langle S_i^\alpha(0)S_j^\alpha(t)\rangle$) computed from Bethe ansatz show rapid reemergence of $Q = \pi$ antiferromagnetism above the light cone after a local spin flip, where nearest neighbor $\langle S_0^\alpha(t)S_1^\alpha(t)\rangle \to \frac{1}{12} - \frac{\ln 2}{3} \simeq -0.1477...$[30] as $t \to \infty$. At first glance, these results are contradictory; but the difference between $\langle S_0^\alpha(0)S_1^\alpha(t)\rangle$ and $\langle S_0^\alpha(t)S_1^\alpha(t)\rangle$ indicates the new AFM correlations form in a basis orthogonal to the original basis. In other words, the $t \to \infty$ state has zero overlap with the $t = 0$ state, in accord with Anderson's orthogonality catastrophe[31].

This process can be more precisely described as quantum scrambling: the delocalization of quantum information over time[32,33]. Typically such physics is studied via out of time order correlators

(OTOC—see Supplementary Note 6 for details). $G(r, t)$ provides an alternative and more experimentally accessible way to study quantum scrambling, quench dynamics, and quantum thermalization in physical systems.

Heuristic understanding of $G(r, t)$: The $\pi/2$ oscillations inside the quantum wake can be understood heuristically as particle-antiparticle annihilation. In an antiferromagnetic chain, a down spin flipped up creates two spinons, while an up spin flipped down creates two antispinons. These quasiparticles interfere as schematically shown in Fig. 4. Spinons from even neighbor sites interfere constructively and produce a full spin flip, while antispinons from odd neighbor sites interfere destructively and annihilate. Thus $G(r, t)$ oscillates on even sites and $Re[G(r, t)] = 0$ on odd sites.

This heuristic spinon interpretation can explain the temperature evolution of $G(r, t)$ in KCuF$_3$. As temperature increases, the static spin correlations and spin entanglement are suppressed[7], which destroys the coherence of the spinons from neighboring sites as illustrated in Fig. 4b, and the oscillations vanish. This also explains the shift to a ferromagnetic wavefront at high temperatures (Fig. 1h). At low temperatures, the spinons propagate atop a substrate of antiferromagnetic correlations, giving rise to antiferromagnetic oscillating interference patterns. At higher temperatures, the static correlations are mostly gone and so are coherence with neighboring sites (evidenced by the vanishing $Im[G(r, t)]$), so the propagating spinons simply appear as a pair of up-spins hopping through the lattice. In this way, the high temperature quantum wake directly shows spinon quasiparticles—one can "see" them in the data. It is striking that a diffuse high-temperature $S(Q, \omega)$ could yield such a clear quasiparticle signature in $G(r, t)$. This technique could have profound implications for identifying exotic quasiparticles in other magnetic systems, although we caution their signatures and interpretation may differ from spinons, calling for theoretical investigation and predictions especially in higher dimensions.

In conclusion, we have shown using KCuF$_3$ scattering that it is possible to resolve real-time spin dynamics of a local quantum quench via neutron scattering. This reveals details about the quantum dynamics which were not obvious otherwise. First, we are able to directly observe the formation of an orthogonal state within the quantum wake as the light cone scrambles the initial state, leaving behind decaying period-doubled $\pi/2$ oscillations. Second, using the imaginary $G(r, t)$ we observe quantum coherence as revealed by noncommuting observables between spins more than 10 neighbors distant in Fig. 2. This is far longer range "quantumness" than is revealed by entanglement witnesses[7]. Third, the high-temperature $G(r, t)$ shows the spinon quasiparticles visually in the data, without need for theoretical models. Such details are difficult or impossible to see with other techniques. In the same way that diffraction pair distribution functions have aided the study of amorphous materials, $G(r, t)$ could be a helpful technique for highly quantum systems whose dynamics are difficult to model—i.e., where traditional analysis tools break down.

The ability to probe short time and space dynamics of quasiparticles is of key importance to both fundamental quantum mechanics research and technological applications. On the fundamental side, the existence of a quantum wake with quasiperiodic $\pi/2$ oscillations shows behavior not captured by bosonization, which means theorists need to re-tool their analytic methods to understand the short-time dynamics of quantum spin chains. Also, measuring $G(r, t)$ at a well-defined finite temperature may shed light on eigenstate thermalization and quantum scrambling in higher-dimensional systems. On the applications side, $G(r, t)$ is more closely related to the output of current quantum computers and so may provide more direct application of this technology. Also, understanding the short-time behavior of quasiparticles in quantum systems is a crucial step in using them for antiferromagnetic spintronic devices[34] where quantum effects can be significant[35,36], or quantum logic operations in real

technologies. Neutron scattering derived $G(r, t)$ provides key insight into these problems.

## Methods

### Extracting $G(r, t)$ from inelastic neutron scattering

The high-energy scattering data was measured on MAPS at ISIS with phonons subtracted, and low energy (<7 meV) scattering data at high temperatures—where the MAPS data is noisy—was filled in with data measured on SEQUOIA[37] at ORNL's SNS[38]. Both data sets were corrected for the magnetic form factor, and the resulting combined data are shown in Fig. 1.

We then masked the elastic scattering (as it is mostly nonmagnetic incoherent scattering), calculated the negative energy transfer scattering using detailed balance, and computed the Fourier transform of the neutron scattering data in both $Q$ and $\hbar\omega$, yielding spin-spin correlation in real space and time $G(r, t) = \langle S(0) \cdot S_r(t) \rangle$. (Prior to transforming, the high energy MAPS data was interpolated using Astropy Gaussian interpolation[39] to create a uniform grid.)

The short-distance long-time $G(r, t)$ dynamics are governed by the lowest measured energies. In this case, the low energy cutoff was 0.7 meV which means $G(r, t)$ is reliable only up to ~$5 \times 10^{-13}$ s. Further details are given in the Supplementary Note 5. Thus, the long-time dynamics are inaccessible to the current data set. This being said, there is an important visible difference between $KCuF_3$ and the Bethe Ansatz $G(r, t)$ at long times: $KCuF_3$ tends toward antiferromagnetic correlations (odd neighbors fade towards red, even neighbors fade more blue), whereas the Bethe ansatz shows no such trend. This is because $KCuF_3$ is magnetically ordered at 6 K due to interchain couplings, and thus has an infinite-time static magnetic pattern; but the idealized 1D Heisenberg AFM does not. Remarkably, the Van Hove function picks this up even though the elastic line—and thus the Bragg intensity—was not included in the transform.

### Theoretical simulations

The Bethe Ansatz plots were produced from data obtained using the ABACUS algorithm[40] which computes dynamical spin-spin correlation function of integrable models through explicit summation of intermediate state contributions as computed from (algebraic) Bethe Ansatz. Linear spin wave calculations were carried out using SpinW[41].

In Supplementary Notes 2-4, we also consider (i) the $S = 1/2$ ferromagnet using both density matrix renormalization group theory (DMRG) and LSWT, (ii) the quantum $S = 1/2$ Ising spin chain for various anisotropies using perturbation theory, and (iii) the quantum $S = 1/2 XY$ spin chain exact solution.

## Data availability

The data used in this study are available in the ORNL LCF database at https://doi.org/10.13139/ORNLNCCS/1872231.

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

## Acknowledgements

We are thankful to Takeshi Egami for enlightening discussions. The research by P.L. was supported by the Scientific Discovery through Advanced Computing (SciDAC) program funded by the US Department of Energy, Office of Science, Advanced Scientific Computing Research and Basic Energy Sciences, Division of Materials Sciences and Engineering. This research used resources at the Spallation Neutron Source, a DOE Office of Science User Facility operated by the Oak Ridge National Laboratory. JSC acknowledges support from the European Research Council (ERC) under ERC Advanced grant 743032 DYNAMINT. The work by D.A.T. and S.E.N. is supported by the Quantum Science Center (QSC), a National Quantum Information Science Research Center of the U.S. Department of Energy (DOE).

## Author contributions

A.S. and D.A.T. conceived and coordinated the project. A.S. analyzed the neutron data, which was collected by A.S., B.L., S.E.N., M.B.S., and D.A.T. Bethe Ansatz simulations were performed by J.S.C., and the *XY* and ferromagnetic simulations were performed by P.L. The physical interpretation of G(r,t) results was developed by A.S., P.L., J.S.C., and D.A.T. The manuscript was written by A.S., J.S.C., and D.A.T. with input from all coauthors.

## Competing interests

The authors declare no competing interests.
