## [Peer Review File · Nature Communications]

REVIEWER COMMENTS

Reviewer #1 (Remarks to the Author):

The manuscript introduces novel technique, based on extracting data from neutron scattering measurements, to significantly deepen our understanding quantum magnetism. This is certainly timely as quantum effects in Heisenberg (anti)ferromagnets have been amply studied in cold atom realizations (see, e.g., <https://www.science.org/doi/10.1126/science.aau4963>) but are much more difficult to probe in solid state systems on the same footing (e.g., nothing akin to measuring Renyi entropy of arbitrary small subsystems of cold atoms as in <https://www.science.org/doi/10.1126/science.aau4963> is possible in solids).

In this sense, recent Ref. 7 by the same team, where entanglement was extracted from neutron scattering data in the same material as in the present paper, is pathbreaking. Furthermore, the present paper extends possibilities to probe quantumness of antiferromagnets with ability to probe short time and space dynamics of quasiparticles and reveal in the process quantum coherence over many sites (that was not possible in Ref. 7) or quantum discord.

This makes the paper highly suitable for publication in top journals. The paper is well written, but typos should be corrected as in

"As shown in Fig. ?? and Fig. 2 of the main text" in the Supplemental Material. Also, the authors mention connection to quantum computing technologies, but there should be also connection to spintronics where antiferromagnets have recently taken one of the central stages, see e.g. <https://journals.aps.org/prx/abstract/10.1103/PhysRevX.9.041016>. However, in most of spintronics studies it is assumed that one is simply observing classical dynamics of initially Neel state. On the other hand, the authors clearly show that in their case the initial site-alternating Neel

order is transformed by time evolution into oscillating antiferromagnetic correlation termed "quantum wake". In principle, separable (i.e., unentangled) Neel states are difficult to

maintain in the course of time evolution (see, e.g., <https://journals.aps.org/prb/abstract/10.1103/PhysRevB.104.214401>) as quantum superpositions are trying to proliferate and it would be useful to comment on the impact of the present manuscript on spintronic experiments with antiferromagnets where quantum corrections should be present (as suggested by a number of recent experiments that cannot be explained by classical micromagnetic modeling as in PRX above) even at room temperature.

Reviewer #2 (Remarks to the Author):

The manuscript reports the new data analysis of the inelastic neutron scattering study of KCuF_3 , a well-studied 1D Heisenberg chain of spin $1/2$ Cu^{2+} ions. The coherent spinon excitations result in the "quantum wake" in real space.

I found this work interesting. It touches on some aspects that did not receive much attention in previous studies on this compound. However, I would not recommend publishing it in Nature Communications. Some comments are given below:

1. From my understanding, inelastic neutron scattering results are usually fitted with $\langle SS \rangle$, which provides a decent understanding of most systems. In the current study, the authors carried out a Fourier transformation on the neutron data and explained the result with a heuristic understanding, which could be hardly examined by other experimental techniques. Overall, I think this work in its current form will be of limited significance to the field.
2. Figure 3 shows selectively blanked Bethe Ansatz patterns with associated Fourier transformed $G(r, t)$. The real space spin correlations $G(r, t)$ seem to be very sensitive to data analysis, but no resolution function analysis is provided in this work.
3. I found the heuristic picture with spinon and antispinon is thought-provoking but for a classical system governed by the spin-wave theory, where fundamental particles are magnons, it's hard to see if a heuristic understanding with magnons works well. But since the spin-wave excitations are more common, a detailed real-space study on classical spin systems should go first.

It would be better for me if the derivation of $G(r, t)$ could be moved to the main text.

Reviewer #3 (Remarks to the Author):

I have reviewed the manuscript by Scheie et al entitled "Quantum wake dynamics in Heisenberg antiferromagnetic chains". In this work, the authors use neutron scattering data to reconstruct the spin-spin correlation functions in KCuF_3 , which realizes a system of approximately independent 1D Heisenberg chains. Natively, neutron scattering gives access to Fourier-space / frequency

correlations. In principle, Fourier transforming these correlations gives access to real-time / real-space correlations, which have been studied in an enormous amount of theoretical work, as well as cold atoms experiment, trapped ions, and other quantum simulation platforms over the last decade. Yet, achieving this Fourier transform in practice requires high enough quality data and has represented a challenge to condensed-matter systems. The authors implement successfully this approach for the first time to a magnetic material. It allows them to observe key signatures of integrability in the system, in the form of long-lived coherent oscillations at $\pi/2$ wave-vector (in contrast to standard antiferromagnetic correlations which are at π wave-vector). This feature (termed "quantum wake" by the authors) is captured by the exact solution of the Heisenberg chain (the Bethe Ansatz), but not by the semi-classical spin-wave theory. It represents therefore a key feature of this strongly-correlated material, observed in the real-time / real-space spin-spin correlations.

Overall, I found the paper very interesting and well written. The paper brings insights from a real condensed-matter system onto questions which have been mostly confined to quantum simulators until now. It inaugurates a novel way to use neutron-scattering data to probe spin-spin correlations in magnetic materials, and I believe that it will represent an inspiring work to both theoreticians and experimentalists of various fields, at the intersection of condensed-matter physics and quantum simulation.

I have a few remarks on the manuscript.

1) The first paragraph on page 3 is unclear. Why do equal-time correlations depend on time? Do the authors consider the dynamics after a local spin-flip? When they say that the state at infinite time has zero correlation with the initial state, do they mean zero overlap? If yes, I suggest to change "correlation" to "overlap" for clarity.

2) There is typo in the caption in Fig. 3. "Panels b, d and e" -> "Panels b, d and f".

3) In part III of the Supplementary Material, the authors should be more explicit about the Ising limit. I understand from Fig. S2 that they set $J_{xy} = \epsilon J_z$ and vary ϵ between 0 and 1, but it would be better to write it explicitly.

4) In part VI of the Supplementary Material, there is a typo: "As show in Fig. ?? and Fig. 2". I guess it is Fig. 1. Also, the very last sentence is unclear. I do not understand "operators do not commute with the original magnetism."

5) I have a more general observation / suggestion. I think that having a true temperature in the system is indeed a key difference with cold atom systems, and quantum simulators in general. It might allow to explore a kind of quantum correlations which was introduced by Malpetti and Roscilde (Phys. Rev. Lett. 117, 130401 -- 2016), namely the difference between equal-time correlations and static cross-susceptibilities (which are equal in classical systems, but may differ due to Heisenberg uncertainty in quantum systems). Both types of correlations may be reconstructed from dynamic structure factors as used by the authors, but as far as I know have never been probed in any actual system, partly due to the difficulty in estimating the temperature in cold atom systems. This represents an interesting perspective for the kind of analysis performed by the authors, which might bring light onto an intrinsic length scale for quantum correlations with a proper discord-like quantum correlations measure.

Referee Response for “Quantum wake dynamics in Heisenberg antiferromagnetic chains” (NCOMMS-22-16634-T)

June 28, 2022

A preliminary note about the resubmission: changes and updates to the manuscript are noted in **red**.

Reviewer #1

We thank the referee for their positive review of our manuscript. We especially appreciate the assessment that our work “significantly deepen[s] our understanding quantum magnetism”, and is “highly suitable for publication in top journals”. The referee made two comments and suggestions, which we have addressed as follows:

1. “typos should be corrected as in ‘As shown in Fig. ?? and Fig. 2 of the main text’ in the Supplemental Material.”

- We thank the referee for pointing out the typo, which we have fixed.

2. “Also, the authors mention connection to quantum computing technologies, but there should be also connection to spintronics where antiferromagnets have recently taken one of the central stages, see e.g. <https://journals.aps.org/prx/abstract/10.1103/PhysRevX.9.041016>. However, in most of spintronics studies it is assumed that one is simply observing classical dynamics of initially Neel state. On the other hand, the authors clearly show that in their case the initial site-alternating Neel order is transformed by time evolution into oscillating antiferromagnetic correlation termed ”quantum wake”. In principle, separable (i.e., unentangled) Neel state are difficult to maintain in the course of time evolution (see, e.g., <https://journals.aps.org/prb/abstract/10.1103/PhysRevB.104.214401>) as quantum superpositions are trying to proliferate and it would be useful to comment on the impact of the present manuscript on spintronic experiments with antiferromagnets where quantum corrections should be present (as suggested by a number of recent experiments that cannot be explained by classical micromagnetic modeling as in PRX above) even at room temperature. ”

- We appreciate the suggestion that this technique could shed light on problems related to antiferromagnetic spintronics, and that entanglement and coherence are very important there. We have amended the conclusion to read “Also, understanding the short-time behavior of quasiparticles in quantum systems is a crucial step in using them for antiferromagnetic spintronic devices [cite Baltz-RMP-2018] where quantum effects can be significant [cite Gray-PRX-2019,Mondal-PRB-2021], or quantum logic operations in real technologies.”

Reviewer #2

We thank the referee for their efforts in reviewing our manuscript. The referee states “The manuscript reports the new data analysis of the inelastic neutron scattering study of KCuF_3 , a well-studied 1D Heisenberg chain of spin $1/2$ Cu^{2+} ions. The coherent spinon excitations result in the ‘quantum wake’ in real space. I found this work interesting. It touches on some aspects that did not receive much attention in previous studies on this compound. However, I would not recommend publishing it in *Nature Communications*. Some comments are given below:” We thank the referee for the assessment that our work is both interesting and original. To the question of whether it belongs in *Nature Communications*, we address the points directly below.

1. “From my understanding, inelastic neutron scattering results are usually fitted with $\langle SS \rangle$, which provides a decent understanding of most systems. In the current study, the authors carried out a Fourier transformation on the neutron data and explained the result with a heuristic understanding, which could be hardly examined by other experimental techniques. Overall, I think this work in its current form will be of limited significance to the field.”

Here the referee questions the overall significance of the work to the field, given the fact that fitting $\langle S \cdot S \rangle$ usually provides a decent understanding of most systems.

- Response: For most semiclassical spin systems, the referee is correct that fitting the spin-spin correlations to a model provides enough information for detailed understanding. (Indeed, in such cases the real space $G(r, t)$ shown in Fig. 2(h) is rather unenlightening.) However, these situations are not the focus of our work. We anticipate the usefulness of our technique will be found in systems which are (a) highly quantum and not semiclassical, and/or (b) have no model which can capture their behavior. In the case of the former, we demonstrate that $G(r, t)$ reveals inherently quantum-mechanical details which are not obvious otherwise (such as period doubled dynamics in a system intensely studied for over 50 years). In the case of the latter, $G(r, t)$ reveals quantum coherence and high temperature quasiparticles in a totally model-independent way, which we believe will be useful for nontrivial systems (like quantum spin liquids) for which rigorous theoretical models do not exist.
 - An analogous situation is the use of atomic pair distribution functions (PDF) in crystallography. For well-behaved crystal systems, a nuclear diffraction pattern is perfectly sufficient to refine the crystalline structure. However for disordered or amorphous materials, it is often easier to interpret and refine the data when it is Fourier transformed to real space. This technique is now widely used in the community, to the point where diffraction instruments are designed with PDF in mind. The real space and time magnetic signal is much the same: it is not always necessary, but in the case of highly quantum systems—for which conventional analysis tools break down—the different view of the data yields unique insight.
2. “Figure 3 shows selectively blanked Bethe Ansatz patterns with associated Fourier transformed $G(r, t)$. The real space spin correlations $G(r, t)$ seem to be very sensitive to data analysis, but no resolution function analysis is provided in this work.”
 - The exercise in Fig. 3 was to show from whence in reciprocal space and frequency the quantum wake features arise. Resolution functions is irrelevant to this calculation, and it was done with theoretical simulations which do not have resolution broadening. To the referee’s direct point however, the top two rows of Fig. 2 show the spin correlations $G(r, t)$ are surprisingly insensitive to data analysis. We expected that the 6 K data would not yield a good match to the Bethe Ansatz because we did not have the low-energy data to fill in; but the match with theory is strong. The resolution function does affect the long-time dynamics, which we explore in section V of the supplemental information. A full exploration of this is beyond the scope of this work because the resolution of direct geometry time of flight neutron spectrometers is extremely nontrivial [see Lin et al, Review of Scientific Instruments 93, 025101 (2022); <https://doi.org/10.1063/5.0079031>]. Fortunately for short times (the focus of this work), resolution effects can be safely set aside.
 3. “I found the heuristic picture with spinon and antispinon is thought-provoking but for a classical system governed by the spin-wave theory, where fundamental particles are magnons, its hard to see if a heuristic understanding with magnons works well. But since the spin-wave excitations are more common, a detailed real-space study on classical spin systems should go first.”
 - Response: Although the heuristic understanding in terms of spinons does not necessarily translate to magnons, the idea of obtaining such real-space real-time correlations is general and applicable also to other spectroscopies and to electronic strongly correlated systems. We explore this with a follow-up study which is available as a preprint: <https://arxiv.org/abs/2203.06332> .
 - Furthermore, as we explain above, we do not expect this technique to be very useful for situations where we already have excellent theoretical analysis tools. Indeed, Figure 2 of the main text shows a fairly uninteresting (and almost entirely real-valued) response from the linear spin wave calculation. The key point is that there are two trajectories for right and left moving spinwaves.

These do not scramble the intervening state, and so are merely spin deviations from an overall antiferromagnetic state. That being said, there are some situations where viewing magnons in real space could be helpful, e.g. magnon bound states [Fukuhara et al, Nature 502, 7679 (2013). <https://doi.org/10.1038/nature12541>], but this is precisely where linear spin wave theory fails, requiring a different approach.

- With respect to the fully classical system, the 1D Heisenberg spin chain real-space dynamics are actually a matter of intense controversy. Specifically, the numerical studies disagree whether the dynamics are diffusive [see Nianbei et al, PRE (2019) 10.1103/PhysRevE.100.062104], [Bagchi, PRB (2013) 10.1103/PhysRevB.87.075133] or Kardar-Parisi-Zhang [see Das et al, PRL (2018) 10.1103/PhysRevLett.121.024101], [Das et al, PRE (2019) 10.1103/PhysRevE.100.042116]. In an early version of this paper we attempted to simulate the classical case via Landau-Lifshitz dynamics, but quickly found that a rigorous simulation of the soliton dynamics requires huge system sizes and were prohibitively time-consuming. We thus were unable to resolve the disputes, and chose to focus on the quantum and semiclassical situation where the physics (strangely) is less controversial.
4. “It would be better for me if the derivation of $G(r, t)$ could be moved to the main text.”
- Response: We agree, and have moved the derivation at the beginning of the supplemental information to the main text.

Reviewer #3

We thank the referee for their thorough review of the paper. The referee writes, “Overall, I found the paper very interesting and well written. The paper brings insights from a real condensed-matter system onto questions which have been mostly confined to quantum simulators until now. It inaugurates a novel way to use neutron-scattering data to probe spin-spin correlations in magnetic materials, and I believe that it will represent an inspiring work to both theoreticians and experimentalists of various fields, at the intersection of condensed-matter physics and quantum simulation.” We are grateful for the positive assessment of our work. Below we respond to the numbered remarks on the manuscript:

1. “The first paragraph on page 3 is unclear. Why do equal-time correlations depend on time? Do the authors consider the dynamics after a local spin-flip? When they say that the state at infinite time has zero correlation with the initial state, do they mean zero overlap? If yes, I suggest to change ‘correlation’ to ‘overlap’ for clarity.”
 - Response: Equal-time correlations depend on time because they measure dynamics after a local spin flip. For clarity we now state this in the paragraph.
 - When we say the infinite time has zero correlations with the initial state, we mean that the expectation value of the dot product is zero. One way of saying this is “overlap”, and so we change the main text to say “overlap” instead of “correlations”.
2. “There is typo in the caption in Fig. 3. ‘Panels b, d and e’ → ‘Panels b, d and f.’”
 - Response: We thank the referee for pointing this out. The caption to Fig. 3 has been fixed.
3. “In part III of the Supplementary Material, the authors should be more explicit about the Ising limit. I understand from Fig. S2 that they set $J_{xy} = \epsilon J_z$ and vary ϵ between 0 and 1, but it would be better to write it explicitly.”
 - Response: We appreciate the suggestion, and have added text to part III of the supplemental material clarifying the model and writing the Hamiltonian explicitly.
4. “In part VI of the Supplementary Material, there is a typo: ‘As show in Fig. ?? and Fig. 2’. I guess it is Fig. 1. Also, the very last sentence is unclear. I do not understand ”operators do not commute with the original magnetism.””
 - The typo has been fixed, and the proper reference added.

- We have updated the last sentence to read “where time-like separated spin operators do not commute with the spin operators at $t = 0$.”
5. “I have a more general observation / suggestion. I think that having a true temperature in the system is indeed a key difference with cold atom systems, and quantum simulators in general. It might allow to explore a kind of quantum correlations which was introduced by Malpetti and Roscilde (Phys. Rev. Lett. 117, 130401 – 2016), namely the difference between equal-time correlations and static cross-susceptibilities (which are equal in classical systems, but may differ due to Heisenberg uncertainty in quantum systems). Both types of correlations may be reconstructed from dynamic structure factors as used by the authors, but as far as I know have never been probed in any actual system, partly due to the difficulty in estimating the temperature in cold atom systems. This represents an interesting perspective for the kind of analysis performed by the authors, which might bring light onto an intrinsic length scale for quantum correlations with a proper discord-like quantum correlations measure.”
- Response: we thank the referee for this observation and suggestion. This will require some careful thought to implement properly (e.g., extracting the imaginary-time integral as in Malpetti eq. 1 from our real-time data), and is therefore beyond the scope of this study. Nevertheless we are grateful for the suggestion, and intend to pursue the idea.

REVIEWERS' COMMENTS

Reviewer #1 (Remarks to the Author):

All three referee reports agree that manuscript offers novel insight into quantum dynamics, including entanglement, of solid state magnetic materials that are usually treated as systems of classical spins in applied disciplines (like spintronics) and have not been explored as quantum systems due to technical difficulties. The paper is further improved (including corrected typos) through the response to the referees and it can be published now as it is.

Regarding the request of the second referee to add some comparison with classical treatment via the Landau-Lifshitz-Gilbert (LLG) equation, the authors correctly point out that this would not be helpful in the context of the present manuscript. It has been also realized very recently that LLG description of nonequilibrium excitations (by current injection) of ferromagnets and antiferromagnets, as the standard approach in spintronics and magnonics, can be highly inappropriate in limits where many-body effects kick in, see e.g.: <https://doi.org/10.1103/PhysRevLett.126.197202> and <https://doi.org/10.1103/PhysRevX.11.021062>.

Reviewer #2 (Remarks to the Author):

I appreciate the authors' detailed response to my comments. The discussion about the limitations of the approach (previous comments 1&3) is very intriguing. Again, I think this work is interesting and well-prepared. It is recommended to be published by Nature Communications after further revision. Given that this may be the first time such an approach is implemented for neutron scattering results on quantum magnets, the range of the systems suitable for this analysis should be mentioned in the main text.

In the response, the authors state that "We anticipate the usefulness of our technique will be found in systems which are (a) highly quantum and not semiclassical, and/or (b) have no model which can capture their behavior. In the case of the former, we demonstrate that $G(r, t)$ reveals inherently quantum mechanical details which are not obvious otherwise (such as period doubled dynamics in a system intensely studied for over 50 years). In the case of the latter, $G(r, t)$ reveals quantum coherence and high temperature quasiparticles in a totally model-independent way, which we believe will be useful for nontrivial systems (like quantum spin liquids) for which rigorous theoretical models do not exist." In the case of quantum spin liquids in 2D or 3D, most candidates are under

debates (Herbertsmithites, YbMgGaO₄, RuCl₃, ...). And since most of them are highly disordered, $G(r,t)$ analysis will be challenging and it is unlikely that the results will significantly improve the understanding. Also, 2D and 3D QSLs are fundamentally different from 1D cases. In the case of systems that have no model which can capture their behavior, it might be skeptical to define quasiparticles and report their behaviors according to $G(r,t)$. In this sense, I assume the main battlefield of $G(r,t)$ is still 1D quantum magnets since the basic excitation is known.

The effect of the resolution is recommended to be mentioned in the main text.

Reviewer #3 (Remarks to the Author):

The authors provided satisfactory answers to the Reviewers; I recommend publication of the manuscript in its present form.

Referee Response for “Quantum wake dynamics in Heisenberg antiferromagnetic chains” (NCOMMS-22-16634A), second response

August 22, 2022

A preliminary note about the resubmission: changes and updates to the manuscript are noted in **red**.

Reviewer #1

“All three referee reports agree that manuscript offers novel insight into quantum dynamics, including entanglement, of solid state magnetic materials that are usually treated as systems of classical spins in applied disciplines (like spintronics) and have not been explored as quantum systems due to technical difficulties. The paper is further improved (including corrected typos) through the response to the referees and it can be published now as it is.

“Regarding the request of the second referee to add some comparison with classical treatment via the Landau-Lifshitz-Gilbert (LLG) equation, the authors correctly point out that this would not be helpful in the context of the present manuscript. It has been also realized very recently that LLG description of nonequilibrium excitations (by current injection) of ferromagnets and antiferromagnets, as the standard approach in spintronics and magnonics, can be highly inappropriate in limits where many-body effects kick in, see e.g.: <https://doi.org/10.1103/PhysRevLett.126.197202> and <https://doi.org/10.1103/PhysRevX.11.021062>.”

- We thank the referee for the positive assessment of our revisions and the endorsement for publication. We are also grateful for the references on the limitations of LLG dynamics in the many-body context.

Reviewer #2

We thank the referee for again reviewing our manuscript. They made several final comments which we address as follows:

“I appreciate the authors’ detailed response to my comments. The discussion about the limitations of the approach (previous comments 1&3) is very intriguing. Again, I think this work is interesting and well-prepared. It is recommended to be published by Nature Communications after further revision. Given that this may be the first time such an approach is implemented for neutron scattering results on quantum magnets, the range of the systems suitable for this analysis should be mentioned in the main text.”

- We thank the referee for the endorsement of our work and recommendation for publication in Nature Communications.
- On the range of systems suitable for this analysis: We have the last sentence before the conclusion to say “... although we caution their signatures and interpretation may differ from spinons, calling for theoretical investigation and predictions especially in higher dimensions.” We also added this sentence to the conclusion: “In the same way that diffraction pair distribution functions have aided the study of amorphous materials, $G(r, t)$ could be a helpful technique for highly quantum systems whose dynamics are difficult to model—i.e., where traditional analysis tools break down.”

“In the response, the authors state that ‘We anticipate the usefulness of our technique will be found in systems which are (a) highly quantum and not semiclassical, and/or (b) have no model which can capture their behavior. In the case of the former, we demonstrate that $G(r, t)$ reveals inherently quantum mechanical details which are not obvious otherwise (such as period doubled dynamics in a system intensely studied for

over 50 years). In the case of the latter, $G(r, t)$ reveals quantum coherence and high temperature quasiparticles in a totally model-independent way, which we believe will be useful for nontrivial systems (like quantum spin liquids) for which rigorous theoretical models do not exist.’ In the case of quantum spin liquids in 2D or 3D, most candidates are under debates (Herbertsmithites, YbMgGaO₄, RuCl₃, ...). And since most of them are highly disordered, $G(r, t)$ analysis will be challenging and it is unlikely that the results will significantly improve the understanding. Also, 2D and 3D QSLs are fundamentally different from 1D cases. In the case of systems that have no model which can capture their behavior, it might be skeptical to define quasiparticles and report their behaviors according to $G(r, t)$. In this sense, I assume the main battlefield of $G(r, t)$ is still 1D quantum magnets since the basic excitation is known.”

- On the question of disordered QSL candidates: We agree that $G(r, t)$ is probably not too useful for a heavily (chemically, not magnetically) disordered system. However, there is a strong push towards synthesizing better and better samples, so one might expect $G(r, t)$ to be able to provide more insight also into these systems in the future. Even with today’s imperfect samples, $Im[G(r, t)]$ might be able to diagnose the degree to which a quantum description is necessary.
- On the question of “defining quasiparticles and reporting their behaviors according to $G(r, t)$ ”: If the referee by this means measuring $G(r, t)$, seeing some geometrical pattern, and using that as basis to conclude it is described by quasiparticles of some description, then we certainly agree that such findings have not been demonstrated to be a sufficient condition for a quasiparticle description. In particular it is not clear what $G(r, t)$ might look like in a system without quasiparticles. Nevertheless, we do believe that $G(r, t)$ will be at least helpful information in identifying exotic magnetic quasiparticles, and in some cases may be all we have to go on. We hope our paper will inspire theoretical predictions of signatures for other quasiparticles to help guide the interpretation of future experiments.
- On the question of higher dimensional QSL materials: It is difficult to say before the fact whether $G(r, t)$ will be useful in higher dimensions, but this is an avenue we intend to explore.

“The effect of the resolution is recommended to be mentioned in the main text.”

- We have added the following sentence to the results section of the main text: **Experimental resolution broadening does affect the long-time $G(r, t)$ dynamics (see Supplemental Information), but the short-time spin correlations $G(r, t)$ are surprisingly insensitive to experimental resolution effects.**

Reviewer #3

The referee made a single sentence comment: “The authors provided satisfactory answers to the Reviewers; I recommend publication of the manuscript in its present form.” We are grateful for the referee’s time reviewing our manuscript, and the endorsement of our paper for publication.